# The Association Between Chronic Heart Failure and Metabolic Syndrome Increases the Cost of Hospitalization

**DOI:** 10.3390/healthcare13111239

**Published:** 2025-05-24

**Authors:** Alexandra Mincă, Claudiu C. Popescu, Dragoș I. Mincă, Amalia L. Călinoiu, Ana Ciobanu, Valeriu Gheorghiță, Dana G. Mincă

**Affiliations:** 1Public Health and Management Department, “Carol Davila” University of Medicine and Pharmacy, 020021 Bucharest, Romania; alexandra.tintea@drd.umfcd.ro (A.M.); dana.minca@umfcd.ro (D.G.M.); 2“Prof. Dr. Agrippa Ionescu” Emergency Clinical Hospital, 011356 Bucharest, Romania; 3Rheumatology Department, “Carol Davila” University of Medicine and Pharmacy, 020021 Bucharest, Romania; 4Cardio-Thoracic Surgery Department, “Carol Davila” University of Medicine and Pharmacy, 020021 Bucharest, Romania; 5Infectious Diseases Department, “Carol Davila” University of Medicine and Pharmacy, 020021 Bucharest, Romania

**Keywords:** cost, chronic heart failure, metabolic syndrome, hospitalization

## Abstract

**Background/Objectives**: This study aimed to observe and compare the real-world total hospitalization costs for patients with chronic heart failure (CHF) and metabolic syndrome (MetS) in Romania. **Methods**: Data were electronically retrieved from three different internal medicine departments of university hospitals in Bucharest, Romania, including all admissions from December 2023 to June 2024. The collected data included demographics, the cost of hospitalization (EUR), and discharge diagnoses (ICD10 codes were used to calculate the Charleston Comorbidity Index, CCI, and to define a surrogate measure for MetS). **Results**: The database query retrieved 4732 hospitalizations (median duration: 4 days; median cost: EUR 1002) of unique patients (53.9% women, average age = 68.7 years), of whom 48.0% had CHF and 11.0% were classified as having MetS. The median hospitalization duration and costs were similar for men and women, despite women being significantly older and having a higher prevalence of CHF. Patients with CHF or MetS were significantly older, had more comorbidities (CCI), and had a higher median hospitalization duration, total hospitalization cost, and cost/day than those without CHF or MetS. The total cost of hospitalization increased steadily from a minimum in patients without CHF or MetS to a maximum in patients with both conditions. **Conclusions**: CHF was highly prevalent among patients admitted to internal medicine wards and was more prevalent among hospitalized women. However, the hospitalization costs did not differ significantly between the sexes. CHF and MetS incrementally increased the total hospitalization costs in these DRG-based reimbursement systems.

## 1. Introduction

Chronic heart failure (CHF) is a progressive clinical syndrome that arises most commonly from structural or functional cardiac abnormalities. Pathophysiologically, CHF involves neurohormonal activation, leading to maladaptive compensatory mechanisms such as sympathetic nervous system overactivation and renin–angiotensin–aldosterone system upregulation [1,2]. Clinically, CHF manifests as dyspnea, fatigue, fluid overload, and exercise intolerance, significantly impairing quality of life. Management strategies include pharmacological interventions along with non-pharmacological measures such as lifestyle modifications and device-based therapies [3,4,5,6]. Despite advances in treatment, CHF remains a major cause of morbidity and mortality worldwide, necessitating ongoing research into novel therapeutic approaches [7,8].

Consequently, CHF entails a substantial economic burden on healthcare systems worldwide due to its high prevalence, frequent hospitalizations, and long-term management requirements [7,9,10,11,12]. The associated costs are primarily driven by hospital admissions [13], pharmacological therapies, outpatient care, and the need for advanced interventions such as implantable devices [14] or heart transplantation [15]. Indirect costs, including loss of productivity and caregiver burden, further contribute to the financial impact. In developed countries, CHF accounts for a significant proportion of total healthcare expenditures [16], with costs projected to rise due to aging populations and increasing disease prevalence [17]. Strategies to reduce the economic burden of CHF include optimizing medical therapy, promoting early diagnosis, and implementing preventive measures to reduce hospital readmissions and slow disease progression [18,19,20].

Metabolic syndrome (MetS) has a high prevalence among CHF patients [21]. The International Diabetes Foundation (IDF) defines MetS as the co-occurrence of central obesity and any two of the following: elevated triglycerides, reduced HDL cholesterol, elevated blood pressure, and elevated fasting plasma glucose [22]. The interplay between MetS and CHF is multifaceted, involving both the contribution of MetS to CHF development and its influence on patient outcomes. MetS may contribute to the risk of developing CHF, particularly through mechanisms such as visceral adiposity [23]. Additionally, CHF and MetS share biomarkers and pathways related to obesity, lipid metabolism, and chronic inflammation [24,25,26]. In this context, their association has been rightly termed cardiometabolic syndrome [27,28], as opposed to cardiorenal syndrome [29,30]. Furthermore, the presence of MetS may have negative prognostic implications for CHF patients [31,32], an observation that requires further investigation. Finally, MetS significantly increases healthcare costs [33].

In this context, the current study aimed to observe and compare the real-world total costs of hospitalization for patients with CHF and MetS in an upper-middle-income European country.

## 2. Materials and Methods

### 2.1. Data

Data were electronically retrieved from three different internal medicine departments of university hospitals in Bucharest, Romania, covering an admission timeframe from December 2023 to June 2024. All patients admitted to the hospitals during the specified period were included, and the first admission form of each unique patient was retained for analysis. On admission, all patients provided written informed consent for the scientific use of their personal and medical data.

Collected data included sex, age, admission and discharge dates, hospitalization costs (reported at an approximate exchange rate of RON 5 to EUR 1), and discharge diagnoses coded with the 10th version of the International Classification of Diseases (ICD-10). Due to the retrospective nature of the study and missing data, quantitative measurement and retrieval of defining criteria (laboratory and anthropometric data) were not possible. ICD-10 codes were used to identify diagnoses included in the Charleston Comorbidity Index (CCI; Table 1) [34] and to define MetS using a proxy strategy—in particular, MetS was defined as either E66 + I10 + E11, E66 + I10 + E78.0-5, or E66 + E11 + E78.0-5—an approach commonly used and validated in retrospective and claims-based research [35,36]. The study was conducted in accordance with the Declaration of Helsinki and approved by the Ethics Committee of “Prof. Dr. Agrippa Ionescu” Clinical Hospital (protocol code 292318/18 April 2024).

### 2.2. Hospitalization Costs

Since 2003, public hospital financing in Romania has been primarily managed through a system based on Diagnosis-Related Groups (DRGs). Initially, the Australian Refined DRG (AR-DRG) version 5 was used for hospital reimbursement [37]; however, in 2010, Romania developed its own DRG variant (RO-DRG). This version incorporated new definitions for comorbidities and complications and adjusted grouping limits for certain cases. This system undergoes periodic updates to enhance its applicability to the Romanian healthcare context. The financial mechanism of hospital reimbursement consists of prospective payments using a mix of methods, including DRG-based case payments, day tariffs, lump sums for national health programs, and fee-for-service payments for outpatient services. The DRG component is central, determining reimbursement based on case complexity and resource requirements. Cost data were derived from the hospital billing system and include the DRG base rate as well as additional charges related to procedures, medications, laboratory tests, imaging, and other ancillary services rendered during hospitalization. As a result, the cost data reflect only direct in-hospital expenditures, excluding outpatient management and indirect costs.

### 2.3. Statistics

Since there were no missing data for the primary variables—mandatory for hospital reimbursement and routinely collected in a standardized format—no data were excluded or imputed. Data distribution normality was assessed using descriptive statistics, normality tests, stem-and-leaf plots, and the Lilliefors-corrected Kolmogorov–Smirnov test. Continuous variables are reported as the “mean ± standard deviation” (SD) if normally distributed or “median (interquartile range)” (IQR) if non-normally distributed. Nominal variables are reported as an “absolute frequency (percentage of group or subgroup)”. Differences in continuous variables among subgroups were assessed using independent-sample *t* tests (for normally distributed continuous variables across groups of dichotomous nominal variables) or Mann–Whitney and Kruskal–Wallis tests (for non-normally distributed continuous variables across groups of nominal variables with two or more states). Associations between dichotomous categorical variables were assessed using χ^2^ tests. All analyses were performed using IBM SPSS Statistics for Windows, version 26.0 (IBM Corp., released 2019, Armonk, NY, USA).

## 3. Results

Database query retrieved 4732 admissions of unique patients within the timeframe. Most of the patients included in the lot were women (53.9%), with an average age of 68.7 years (Table 2).

Regarding the diagnoses included in the CCI, CHF was highly prevalent in the sample (48.0%), followed by diabetes mellitus (30.5%), chronic kidney disease (19.8%), myocardial infarction (18.8%), and cirrhosis (17.3%; Table 3). Approximately 11.1% of patients were categorized as having MetS. Within the diagnostic criteria for MetS, arterial hypertension was the most prevalent (50.0%), followed by dyslipidemia (38.5%), diabetes mellitus (30.5%), and obesity (19.8%; Table 3). In terms of time, the median hospitalization duration was 4 days, with a median hospitalization cost of EUR 1002 (Table 2).

Compared to men, women were significantly older (average age of 70.0 ± 13.7 years versus 67.3 ± 13.1 years; *p* < 0.001) and had a significantly higher prevalence of CHF (49.4% versus 46.4%; *p* = 0.042; Figure 1a). However, there were no significant differences in median hospitalization duration (4.0 (6.0) days versus 4.0 (6.0) days; *p* = 0.087), median total hospitalization costs (EUR 1002.4 (7346.2) versus EUR 1002.2 (7337.8); *p* = 0.938), median daily hospitalization costs (EUR 305.1 (1478.9) versus EUR 322.0 (1474.2); *p* = 0.214), or median CCI scores (5.0 (3.0) versus 5.0 (3.0); *p* = 0.635).

Compared to patients without CHF, those with CHF were significantly older (average age of 72.2 ± 11.7 years versus 65.4 ± 14.1 years; *p* < 0.001; Figure 2a), had significantly longer median hospitalization durations (5.0 (7.0) days versus 3.0 (6.0) days; *p* < 0.001; Figure 2b), higher median total hospitalization costs (EUR 1013.8 (7198.1) versus EUR 728.3 (7384.2); *p* < 0.001; Figure 2c), higher daily hospitalization costs (EUR 326.7 (1479.2) versus EUR 312.2 (1476.1); *p* = 0.044), and higher median CCI scores excluding CHF (5.0 (3.0) versus 4.0 (3.0); *p* < 0.001; Figure 2d).

Compared to patients without MetS, those with MetS were more often women (46.9% versus 39.2%; *p* < 0.001), significantly older (68.8 ± 13.7 years versus 67.5 ± 11.1 years; *p* = 0.029), and had a significantly higher prevalence of CHF (55.7% versus 47.1%; *p* = 0.042; Figure 1b). Median hospitalization durations were similar between the groups (4.0 (5.0) days versus 4.0 (6.0) days; *p* = 0.552); however, patients with MetS had significantly higher median total hospitalization costs (EUR 6552.2 (7196.8) versus EUR 951.1 (7350.3); *p* < 0.001), higher median daily hospitalization costs (EUR 905.9 (2500.2) versus EUR 285.2 (1215.3); *p* < 0.001), and higher median CCI scores (5.0 (3.0) versus 4.0 (3.0); *p* < 0.001).

Notably, the total cost of hospitalization increased steadily from a minimum in patients without CHF or MetS to a maximum in patients with both conditions (Figure 3).

## 4. Discussion

### 4.1. Sex Difference in CHF Prevalence

The first observation of the current study was the higher prevalence of CHF among women, who were significantly older than the men. However, hospitalization costs did not differ significantly between the sexes. This difference in CHF prevalence appears to rise in the pre-diagnosis period, as men and women exhibit different risk profiles for CHF [38,39]. Men are more likely to develop CHF with a reduced ejection fraction, whereas women more frequently exhibit a preserved ejection fraction [40,41]. Women with CHF are more likely to experience symptoms such as dyspnea and fatigue, which often prompt them to seek hospitalization, and, consequently, report a lower quality of life [42,43]. In contrast, men may present with more overt signs of volume overload, such as peripheral edema [44,45]. Furthermore, women tend to exhibit a heightened inflammatory response [46], which may not contribute to differences in disease progression and prognosis [47,48]. The diagnosis of CHF in women is often delayed due to atypical symptoms and under-recognition of a preserved ejection fraction [49,50,51]. Despite these differences, women generally have better survival rates than men with CHF [52,53], particularly in cases of a preserved ejection fraction. The reasons for this survival advantage are not fully understood [54] but may be related to differences in ventricular remodeling, hormonal influences, and healthcare-seeking behavior. Social factors, including disparities in access to care and adherence to treatment, may also contribute to sex-based differences in CHF outcomes. Studies have also identified sex-based differences in healthcare costs among patients with CHF. For example, men with CHF have been shown to incur higher healthcare expenses than women [55,56], an observation that was not replicated in the current study. These cost disparities may be attributed to several factors, including differences in disease severity, comorbidities, and treatment approaches between the sexes.

### 4.2. High Frequency of CHF Among Admission Causes

Apart from the difference in prevalence by sex, another observation of the current study was the high prevalence of CHF (48.0%) among patients admitted to internal medicine wards. The literature confirms that CHF is a significant contributor to hospital admissions globally. For example, a nationwide analysis in Thailand between 2008 and 2013 reported that heart failure in adult patients accounted for approximately 1.7% of all adult hospitalizations [57]. Another relevant report from Spain, in which the Basque Health Service analyzed data from 2011 to 2015, revealed that 36% of unplanned admissions were due to heart failure [58]. Apart from the high prevalence of CHF, the study observed a relatively low prevalence of MetS (11.1%) among hospitalized patients. In contrast, the literature consistently reports high prevalence rates among hospitalized patients, regardless of their primary diagnoses or clinical profiles [59,60,61]. This discrepancy is likely due to the under-reporting of relevant ICD codes, which underscores a key weakness of the DRG system: its focus on ICD10 codes that increase a case’s complexity.

### 4.3. The Cost of CHF and MetS

The main observation of this study was the confirmation, within particular medical and financial environments, that both CHF and MetS increase hospitalization costs. CHF increases hospitalization costs through multiple mechanisms, including prolonged hospital stays, frequent readmissions, intensive treatments, the use of specialized care units, management of comorbidities, and end-of-life care needs. A key point raised by the current study is that hospitalizations account for a significant portion of CHF-related healthcare expenditures, with some studies reporting that approximately 75–80% of direct costs for heart failure are attributable to inpatient hospital stays [9]. Therefore, further research aimed at reducing CHF hospitalization costs will likely require a complex approach: preventing hospital admissions through improved outpatient care [62], shortening hospital stays via standardized management protocols, minimizing readmissions through post-discharge monitoring [63], and using advanced therapies and technologies to reduce long-term costs. The observation that both CHF and MetS are associated with higher hospitalization costs highlights the economic burden of cardiometabolic multimorbidity. This emphasizes the need for early detection, coordinated chronic disease management, and preventive care strategies. From a healthcare policy perspective, targeted interventions, such as structured outpatient care programs, improved adherence to guideline-directed therapy, and lifestyle modification support, could reduce hospital utilization and associated costs. Moreover, risk-based resource allocation models or bundled payment systems that reflect comorbidity burden may improve cost efficiency while maintaining quality of care. These issues should be more thoroughly addressed within the specific framework of each national healthcare system, especially in emerging economies, where reducing unnecessary costs may improve access to medical care.

### 4.4. Policy Implications of Observed CHF Costs

This study contributes to current knowledge by providing real-world, administrative cost data on CHF and MetS from an upper-middle-income European country, based on a comprehensive and mandatory dataset for reimbursement. Unlike many prior studies relying on clinical trial populations or limited regional samples, this analysis reflects the actual burden of routine hospital care and highlights the additive economic impact of metabolic comorbidity in CHF. These findings can inform health policy by identifying high-cost patient subgroups for targeted interventions and supporting cost-containment strategies such as preventive care, integrated chronic disease management, or value-based reimbursement models. Potential implications for Romanian health policy and DRG system refinement include costs associated with cardiometabolic multimorbidity and incorporating comorbidity clusters such as MetS into DRG weighting or case-mix adjustment models. For other countries with similar hospital coding and financing systems, our approach demonstrates how existing administrative data can be leveraged for health services research and economic surveillance. The implications go beyond cost awareness, pointing toward actionable opportunities for prevention and system-level optimization.

### 4.5. Limitations of the Study

There are several limitations of this study that may impact the interpretation of its results. Regarding design-based limitations, first, the descriptive retrospective design does not establish cause-and-effect relationships between CHF and hospitalization costs; for example, higher costs may be due to disease severity, comorbidities, or hospital policies. Additionally, this design does not allow for the evaluation of long-term outcomes (e.g., readmission costs, medication adherence, home healthcare, and rehabilitation expenses). This is important, as CHF patients accrue costs throughout the course of the disease, and lifetime costs are equally as important for a complete evaluation of economic impact. Second, indirect costs such as productivity losses due to morbidity and premature mortality (e.g., absenteeism, job loss, early retirement, reduced work capacity, and the burden on informal caregivers) were not recorded, likely underestimating the total economic impact. Third, selection bias may have limited the study’s ability to capture all CHF patients, especially those treated in outpatient settings or those who avoid hospitalization due to financial or geographic barriers. While outpatient settings typically generate less cost than admission wards, prescribed medications and diagnostic imaging may still produce significant overall costs. Fourth, as data were collected from a single city, findings may not be generalizable to other healthcare settings or populations. Therefore, a multi-centric or multi-national study design would be better suited. In terms of data-based limitations, first, the use of discharge ICD-10 codes to define clinical conditions—an inherent constraint in retrospective, administrative data-based studies—may be subject to misclassification bias, as coding accuracy depends on documentation quality, coder expertise, and institutional practices. Second, the analysis was limited by the absence of detailed clinical and socioeconomic variables, which precluded adjustment for potential confounders. Third, because the study relied on hospital records, errors in coding or missing data may have affected the accuracy of cost estimates.

## 5. Conclusions

CHF is highly prevalent (48.0%) among patients admitted to internal medicine wards and is more prevalent among hospitalized women, without significant differences in hospitalization costs compared to men. CHF and MetS were associated with incrementally increases in total hospitalization costs in DRG-based reimbursement systems. Addressing CHF-related costs requires prospective cost-reduction strategies focused on early disease management, optimized outpatient care, and the prevention of avoidable hospital admissions.

## Figures and Tables

**Figure 1 healthcare-13-01239-f001:**
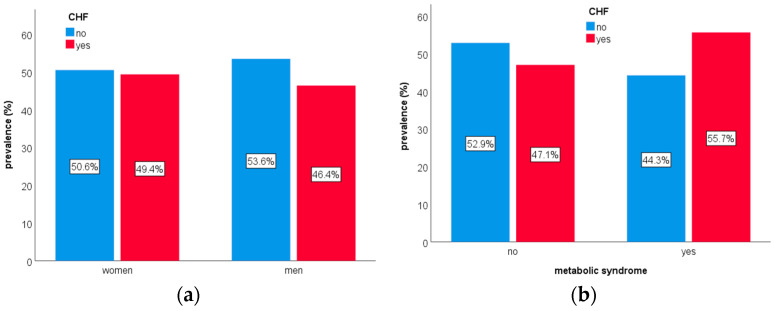
The prevalence of chronic heart failure (CHF) by sex ((**a**): 2552 women and 2180 men; χ^2^ test, *p* = 0.042) and among patients with or without metabolic syndrome ((**b**): n = 526 and n = 4206, respectively; χ^2^ test, *p* < 0.001).

**Figure 2 healthcare-13-01239-f002:**
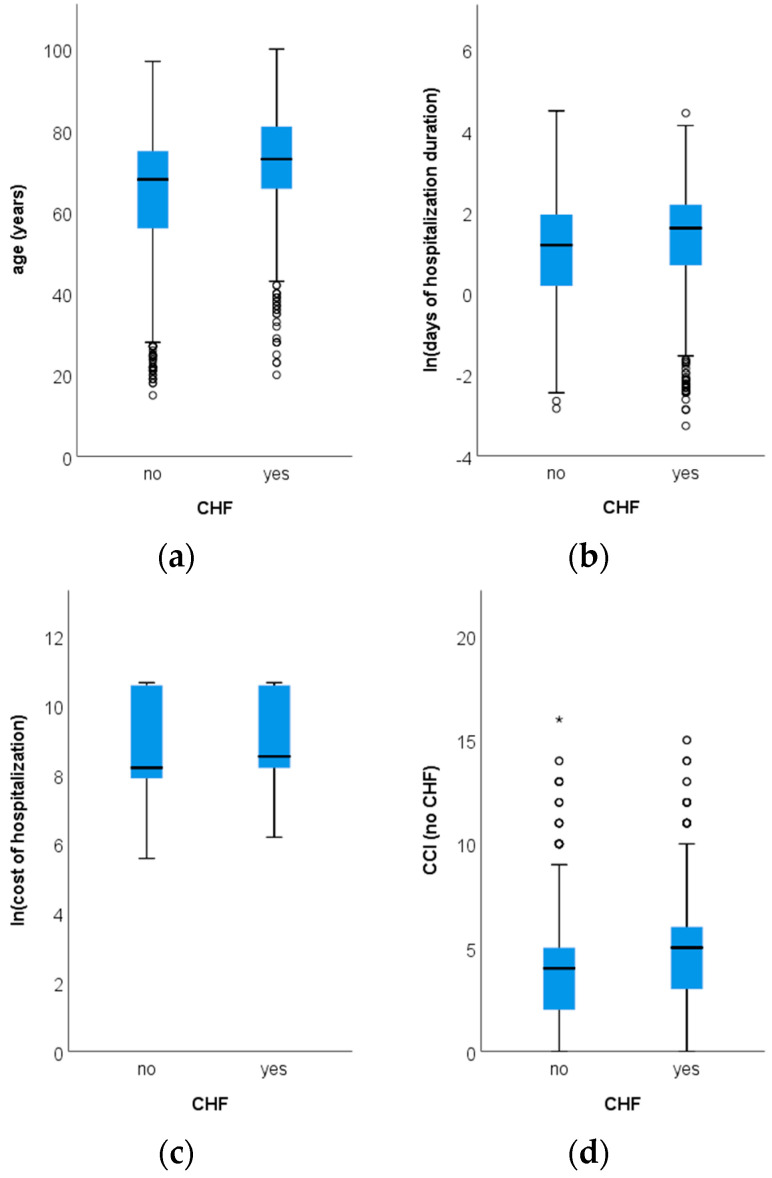
Differences among patients with or without chronic heart failure (CHF) in terms of age ((**a**); *p* = 0.042), hospitalization duration ((**b**); *p* < 0.001), hospitalization cost ((**c**); *p* < 0.001), and CCI (Charleston Comorbidity Index) score excluding CHF ((**d**); *p* < 0.001; circles and * represent outlier values). Note: all differences are evaluated using Mann–Whitney tests; hospitalization duration and cost of hospitalization are reported as natural logarithms.

**Figure 3 healthcare-13-01239-f003:**
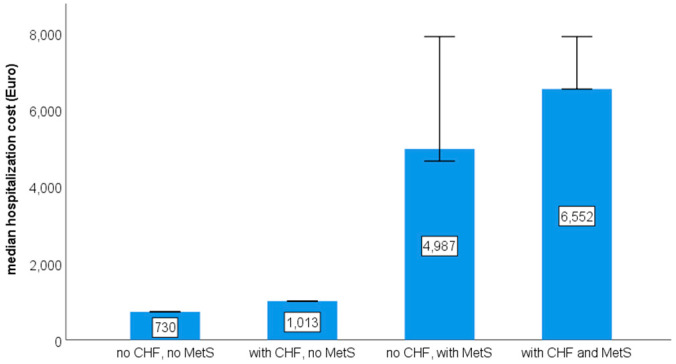
Bar chart with error bars (95% confidence intervals) reporting median total hospitalization cost (EUR) in patients with or without chronic heart failure (CHF), metabolic syndrome (MetS), and their combinations (Kruskal–Wallis, H = 162 (3), *p* < 0.001).

**Table 1 healthcare-13-01239-t001:** ICD10 codes for the studied diagnoses.

Diagnosis	ICD10 Code
chronic heart failure	I50
myocardial infarction	I21, I25
peripheral vascular disease	I73
cerebrovascular disease	I60, I61, I63, I64, I67, G45.8, G45.9
dementia	F00, F01, F03
chronic pulmonary disease	J44
rheumatic disease	M05, M06, M45, L40.5, M07, M32, M33, M34, M35
peptic ulcer disease	K27
cirrhosis	K70, K71, K74, K76
variceal bleeding	I85.0, I98.3
diabetes mellitus	E10, E11, E12, E13, E14
hemiplegia	G81
chronic kidney disease	N18
solid tumor	C
metastatic cancer	C78, C79, C80
leukemia	C90, C91, C92, C93, C94, C95
lymphoma	C81, C82, C83, C84, C85, C86, C88
AIDS	B20, B21, B22, B23, B24
obesity	E66
arterial hypertension	I10
dyslipidemia	E78

Abbreviations: AIDS—Acquired Immunodeficiency Syndrome; ICD10—International Classification of Diseases version 10.

**Table 2 healthcare-13-01239-t002:** General characteristics (n = 4732).

Variable	Observed
women	53.9%
age (years, average ± SD)	68.7 ± 13.4
hospitalization duration (days, median (IQR))	4.0 (5.9)
total cost of hospitalization (EUR, median (IQR))	1002.1 (7338.3)
total cost of hospitalization (RON, median (IQR))	5010.2 (36,690.1)
cost per day of hospitalization (EUR, median (IQR))	322.2 (1476.8)
cost per day of hospitalization (RON, median (IQR))	1611.2 (7347.9)
Charlson Comorbidity Index (average ± SD)	4.8 ± 2.5

Notes: IQR—interquartile range; SD—standard deviation.

**Table 3 healthcare-13-01239-t003:** Prevalence of diagnoses included in the CCI and metabolic syndrome definition (n = 4732).

Diagnosis	Prevalence	Diagnosis	Prevalence
chronic heart failure	48.0%	hemiplegia	0.4%
myocardial infarction	18.8%	chronic kidney disease	19.8%
peripheral vascular disease	0.9%	solid tumor	8.7%
cerebrovascular disease	6.5%	metastatic cancer	2.8%
dementia	4.8%	leukemia	0.5%
chronic pulmonary disease	9.6%	lymphoma	0.3%
rheumatic disease	2.9%	AIDS	0.1%
peptic ulcer disease	0.0%	obesity	19.8%
cirrhosis	17.3%	arterial hypertension	50.0%
variceal bleeding	0.0%	dyslipidemia	38.5%
diabetes mellitus	30.5%	metabolic syndrome	11.1%

Abbreviations: AIDS—Acquired Immunodeficiency Syndrome; CCI—Charlson Comorbidity Index.

## Data Availability

The raw data supporting the conclusions of this article will be made available by the authors on request.

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
