# Peer review of "The Association Between Chronic Heart Failure and Metabolic Syndrome Increases the Cost of Hospitalization"

_healthcare, 2025, doi:10.3390/healthcare13111239_

Round 1
Reviewer 1 Report
Comments and Suggestions for Authors
This study, although the results are logical, is nevertheless of value in understanding the economic burden associated with chronic heart failure (CHF) and metabolic syndrome (MS).
Relevance of the topic - the study addresses the important issue of the relationship between CHF and MS and its impact on the cost of hospitalisation.
The representativeness of the sample is good - analysis of data from more than 4700 hospitalisations provides sufficient statistical power for the study.
Limitations of the study have been mentioned by the authors but need to be described in more detail.
Overall, the article is a qualitative study that expands the understanding of the economic aspects of treating patients with with CHF and MS. The results highlight the need to develop strategies aimed at reducing the frequency of hospitalisations and optimising the cost of treatment for this category of patients.
Author Response
Comment 1.1: Limitations of the study have been mentioned by the authors but need to be described in more detail.
Response 1.1: Thank you very much for pointing this out. The limitations have been expanded (with tracked changes) for a better communication of the studies’ limits.
Reviewer 2 Report
Comments and Suggestions for Authors
The manuscript titled “The association between chronic heart failure and metabolic syndrome increases the cost of hospitalization” by A. Minca et al. addresses an important topic, providing data from their country, which may differ from those of Western Europe and other regions. The manuscript is generally well written and follows appropriate scientific style.
A few comments to improve the manuscript before publication:
- Lines 49–50: Please update the reference to the current guidelines and clarify the recommendations regarding medications.
- Table 2: It is unnecessary to present data for bith sexes in this table. Additionally, remove the statistical abbreviation explanations if they are already defined in the statistic section.
Good luck!
Author Response
Comment 2.1: Lines 49–50: Please update the reference to the current guidelines and clarify the recommendations regarding medications.
Response 2.1: Thank you for pointing this out. The guideline reference has been updated, and recommendations for medications have been added.
Comment 2.2: Table 2: It is unnecessary to present data for both sexes in this table. Additionally, remove the statistical abbreviation explanations if they are already defined in the statistic section.
Response 2.2: Yes, reporting both sexes and already-explained statistical abbreviation explanations is redundant, we have removed it, thank you.
Reviewer 3 Report
Comments and Suggestions for Authors
The article seems to be well written, however, there are concerns regarding the novelty of this information. There has been so much information presented in previous articles regarding the burden of congestive heart failure on the healthcare.
The authors have to clearly address the following: how does this manuscript make an impact on the current knowledge in the field? how can this data be used by others? what consequences do these findings make when it comes to the healthcare system in the given country or other places?
Author Response
Comment 3.1: The article seems to be well written, however, there are concerns regarding the novelty of this information. There has been so much information presented in previous articles regarding the burden of congestive heart failure on the healthcare. The authors have to clearly address the following: how does this manuscript make an impact on the current knowledge in the field? How can this data be used by others? What consequences do these findings make when it comes to the healthcare system in the given country or other places?
Response 3.1: Thank you for helping us to improve our manuscript! Indeed, the literature is abundant, but our design is less frequent (in the SLR made for this study, which is under publication, we only included around 20 studies since 2008). Even so, the questions raised the review are very important, therefore a paragraph answering all of them has been added to the Dission section, before the limitations of the study, as sort of a strengths section of the study (text with tracked changes).
Reviewer 4 Report
Comments and Suggestions for Authors
Major Comments
-
Definition of Metabolic Syndrome:
-
The surrogate ICD-10 code-based definition of MetS (E66 + I10 + E11 or similar combinations) lacks validation. Please cite literature justifying this method or include sensitivity analyses.
-
-
Statistical Rigor:
-
The analysis is descriptive. However, given the large sample size and implications for policy, adjustment for confounding via multivariable regression (e.g., linear or gamma regression for costs) or propensity scoring is essential.
-
-
Cost Interpretation:
-
Report costs in both Romanian leu and euros for clarity.
-
Clarify if costs include DRG base rate, procedure-related charges, medications, or ancillary services.
-
-
Misclassification Bias:
-
The use of discharge ICD-10 codes to define both CHF and MetS is prone to misclassification. Please discuss this in the limitations.
-
-
Discussion Redundancy:
-
The discussion is overly lengthy, repeating background content. Focus on study implications, policy relevance, and comparison with similar studies.
-
Minor Comments
-
Abstract:
-
The term "upper-middle-income European country" can be replaced with “Romania” directly for clarity.
-
Remove abbreviations like Met without prior definition.
-
-
Methods:
-
Indicate how missing data were handled (e.g., missing diagnosis codes, cost outliers).
-
Specify the ethics approval reference number in a more visible location, such as Methods or footnote.
-
-
Tables and Figures:
-
Figure 3 would benefit from confidence intervals or error bars.
-
Include legends in figures to improve standalone readability.
-
-
Grammar and Language:
-
Numerous grammatical and punctuation issues (e.g., “CHF and Met” instead of “CHF and MetS”; inconsistent use of semicolons in lists).
-
Suggest full language editing.
-
-
Conclusion:
-
End with a forward-looking statement—e.g., need for prospective cost-reduction strategies in CHF-MetS patients.
-
The manuscript would benefit from careful English language editing to improve clarity, grammar, and sentence structure. Several sections contain awkward phrasing, redundant expressions, and inconsistent terminology (e.g., use of abbreviations like "Met" instead of "MetS"). A professional English editing service or thorough revision by a fluent speaker is recommended to enhance the overall readability and precision of the scientific message.
Author Response
Comment 4.1: Definition of Metabolic Syndrome: The surrogate ICD-10 code-based definition of MetS (E66 + I10 + E11 or similar combinations) lacks validation. Please cite literature justifying this method or include sensitivity analyses.
Response 4.1: This surrogate strategy of defining MetS is a commonly used proxy definition in retrospective or claims-based research. Two examples of such references were inserted in the text to justify this method.
Comment 4.2: Statistical Rigor: The analysis is descriptive. However, given the large sample size and implications for policy, adjustment for confounding via multivariable regression (e.g., linear or gamma regression for costs) or propensity scoring is essential.
Response 4.2: Indeed, it would have been an excellent objective. But this study was based on a minimal dataset containing only age, sex, admission and discharge dates, and ICD-10 codes, which are routinely collected for administrative purposes. As a result, the ability to adjust for potential confounding factors, such as disease severity, socioeconomic status, comorbidities beyond those coded, or treatment details, was limited. While multivariable modeling or propensity score adjustment could enhance causal inference in richer datasets, these methods were not feasible here due to the absence of relevant covariates. A statement on this has been added to the limitations section.
Comment 4.3: Cost Interpretation: Report costs in both Romanian leu and euros for clarity.
Response 4.3: Table 2 has been modified to include costs in Romanian leu.
Comment 4.4: Cost Interpretation: Clarify if costs include DRG base rate, procedure-related charges, medications, or ancillary services.
Response 4.4: Thank you for raising this issue. A clarifying statement has been added to the “2.2. Hospitalization costs” subheading from the Methods section.
Comment 4.5: Misclassification Bias: The use of discharge ICD-10 codes to define both CHF and MetS is prone to misclassification. Please discuss this in the limitations.
Response 4.5: Thank you for pointing this out. A statement on this has been added to the limitations section.
Comment 4.6: Discussion Redundancy: The discussion is overly lengthy, repeating background content. Focus on study implications, policy relevance, and comparison with similar studies.
Response 4.6: The discussion section has 4 paragraphs: the first discusses our observation that CHF is more prevalent in women, the second discusses our observation of the high prevalence of CHF among admitted patients, the third discusses the increase in costs caused by CHF and MetS and the fourth is the limitation section. The first 3 paragraphs all include comparison with similar studies. The discussion on cost (third paragraph), which reflects the objective of the study, has been expanded to reflect policy relevance.
Comment 4.7: The term "upper-middle-income European country" can be replaced with “Romania” directly for clarity.
Response 4.7: Done: „ upper-middle-income European country” -> „Romania”.
Comment 4.8: Remove abbreviations like Met without prior definition.
Response 4.8: Abbreviation management has been rechecked and corrected in the entire manuscript.
Comment 4.9: Methods: Indicate how mis sing data were handled (e.g., missing diagnosis codes, cost outliers).
Response 4.9: The primary extracted data set contains sex, age, date of admission, date of discharge, for ICD-10 codes for admission and discharge diagnoses. All these variables are law-required for reimbursement, so there were no missing data. A statement informing on this has been added to the Statistics section.
Comment 4.10: Specify the ethics approval reference number in a more visible location, such as Methods or footnote.
Response 4.10: Done: ethics approval reference number was added in the Methods, under heading 2.1.
Comment 4.11: Figure 3 would benefit from confidence intervals or error bars.
Response 4.11: Error bars have been added.
Comment 4.11: Include legends in figures to improve standalone readability.
Response 4.11: Figure legends have been expanded to improve standalone readability.
Comment 4.12: Numerous grammatical and punctuation issues (e.g., “CHF and Met” instead of “CHF and MetS”; inconsistent use of semicolons in lists). Suggest full language editing.
Response 4.12: Thorough revision of the text by a fluent speaker has been done, all grammatical and punctuation issues have been addressed.
Comment 4.13: Conclusion: End with a forward-looking statement—e.g., need for prospective cost-reduction strategies in CHF-MetS patients.
Response 4.13: The final phrase of the Conclusion has been updated to reflect a forward-looking statement.
Round 2
Reviewer 3 Report
Comments and Suggestions for Authors
The Authors have made the requested changes
Author Response
We deeply thank the reviewer for the time and effort for improving our manuscript.
Reviewer 4 Report
Comments and Suggestions for Authors
English Language and Style
-
Several grammatical and syntactical errors reduce clarity (e.g., "mMetabolic syndrome", "Thiese", "diseases which that"). A thorough English language edit is needed.
-
Some sentences are overly long and complex; shortening would improve readability.
2. Introduction
-
The background is comprehensive but too lengthy and sometimes redundant. Condense overlapping discussions (e.g., pathophysiology of CHF is repeated in detail).
3. Methods
-
The definition of MetS using ICD-10 codes should be more transparently justified. Cite the specific validation study for this algorithm.
-
Clarify that cost data reflect only direct in-hospital expenditures (exclude outpatient or indirect costs).
-
Provide more clarity on how ICD codes were mapped to CCI (i.e., were any automated tools or algorithms used?).
4. Results
-
Statistical reporting could benefit from consistency:
-
Include p-values for all comparisons mentioned.
-
Use standardized formatting for median (IQR) and mean ± SD.
-
-
Some figures need clearer labeling, particularly the cost data presented in logarithmic form (specify in figure legends).
5. Discussion
-
The interpretation is rich and cites relevant literature, but the section is overly long and could be more concise.
-
It would benefit from clearer subheadings (e.g., “Sex Differences”, “MetS Prevalence”, “Policy Implications”).
-
Add a comment on how findings could influence Romanian health policy or DRG refinement.
6. Limitations
-
Well acknowledged but should better distinguish between design-based limitations (e.g., retrospective design) and data-based limitations (e.g., ICD coding bias).
The manuscript would benefit from a thorough English language edit. While the overall meaning is clear, there are multiple grammatical issues, awkward phrasing, and typographical errors throughout the text (e.g., "mMetabolic syndrome", "which that", "Thiese"). Sentence structures are occasionally overly complex or redundant, which affects readability and clarity. A professional language editing service or review by a native English speaker is recommended to improve flow and ensure consistency.
Author Response
We deeply thank the reviewer for the time, attention and effort in improving our manuscript.
Comment 4.1. Several grammatical and syntactical errors reduce clarity (e.g., "mMetabolic syndrome", "Thiese", "diseases which that"). A thorough English language edit is needed. Some sentences are overly long and complex; shortening would improve readability.
Response 4.1. Thank you for pointing this out. We now have used the professional English editing service provided by the Healthcare journal for this manuscript with the latest modifications. So, this submitted text is now professionally edited for English, as per the attached certificate.
Comment 4.2. The background is comprehensive but too lengthy and sometimes redundant. Condense overlapping discussions (e.g., pathophysiology of CHF is repeated in detail).
Response 4.2. Indeed, this is an economical perspective on CHF, so many of the pathophysiological and clinical considerations have been removed to condense the background and to emphasize the cost aspect of the ensuing research.
Comment 4.3. The definition of MetS using ICD-10 codes should be more transparently justified. Cite the specific validation study for this algorithm.
Response 4.3. We thank the reviewer for this valuable suggestion. Validation of this definition comes from two studies by Hivert and colleagues. In their 2009 study, they developed an EHR-derived case definition for MetS based on NCEP ATP III criteria and validated it by directly measuring MetS components in a subset of patients. The results showed that the ICD-based algorithm had 73% sensitivity and 91% specificity for detecting MetS compared to actual measured criteria, which is good for our intended purposes. In a follow-up study, Hivert et al. applied a similar MetS identification algorithm to a Canadian hospital and examined outcomes. Their previously validated algorithm based on routinely collected ICD data was shown to be useful for risk stratification in a real-world clinical setting. Both of these studies have now been cited in our manuscript (please see references 35 and 36) in order to more transparently justify our approach. The methods now also mentions that the approach was previously validated.
Comment 4.4. Clarify that cost data reflect only direct in-hospital expenditures (exclude outpatient or indirect costs).
Response 4.4. We thank the reviewer for this important suggestion. This is now clearly acknowledged in the Methods, at the end of subheading “2.2. Hospitalization costs”.
Comment 4.5. Provide more clarity on how ICD codes were mapped to CCI (i.e., were any automated tools or algorithms used?).
Response 4.5. We used combined basic functions of Excel. We searched the columns of extracted ICD-10 codes using ad-hoc formulas such as [=IF(SUMPRODUCT(--ISNUMBER(SEARCH("I50", BC2:DO2))) > 0, 1, 0)] to find and return the desired diagnoses (in this example we searched for heart failure). We do not feel that this basic Excel programming needs to be reported, since searching columns is not an original design item. Plus, Table 1 already reports what decisions were made in terms of ICD codes used for these searches.
Comment 4.6. Statistical reporting could benefit from consistency: include p-values for all comparisons mentioned.
Response 4.6. We compared men and women and this is the paragraph in the results: “Compared to men, women were significantly older (average age of 70.0 ± 13.7 years versus 67.3 ± 13.1 years; p < 0.001) and they had a significantly higher prevalence of CHF (49.4% versus 46.4%; p = 0.042; Figure 1A), but similar median hospitalization durations (4 (6) days versus 4 (6) days; p = 0.087), similar median total hospitalization costs (1002 (7346) € versus 1002 (7338) €; p = 0.938), similar median costs per day of hospitalization (305 (1479) € versus 322 (1474) €; p = 0.214) and similar median CCI scores (5 (3) versus 5 (3); p = 0.635).” We compared patients with and without CHF, and this is the paragraph from the results: “Compared to patients without CHF, those with CHF were significantly older (average age of 72.2 ± 11.7 years versus 65.4 ± 14.1 years; p < 0.001; Figure 2A), they had significantly longer median hospitalization durations (5 (7) days versus 3 (6) days; p < 0.001; Figure 2B), significantly higher median total hospitalization costs (1014 (7198) € versus 728 (7384) €; p < 0.001; Figure 2C), significantly higher median costs per day of hospitalization (327 (1479) € versus 312 (1476) €; p = 0.044) and significantly higher median CCI scores without including CHF itself (5 (3) versus 4 (3); p < 0.001; Figure 2D).” We compared patients with and without MetS, and this is the paragraph from the results: “Compared to patients without the MetS, patients with the MetS were more frequently women (46.9% versus 39.2%; p < 0.001), were significantly older (68.8 ± 13.7 years versus 67.5 ± 11.1 years; p = 0.029) and they had a significantly higher prevalence of CHF (55.7% versus 47.1%; p = 0.042; Figure 1B), similar median hospitalization durations (4 (5) days versus 4 (6) days; p = 0.552), but significantly higher median total hospitalization costs (6552 (7197) € versus 951 (7350) €; p < 0.001), significantly higher median costs per day of hospitalization (906 (2500) € versus 285 (1215) €; p < 0.001) and significantly higher median CCI scores (5 (3) versus 4 (3); p < 0.001).” As the manuscript shows, all of these comparisons have reported p values.
Comment 4.7. Statistical reporting could benefit from consistency: Use standardized formatting for median (IQR) and mean ± SD.
Response 4.7. Standardized formatting for median (IQR) and mean ± SD are now used.
Comment 4.8. Some figures need clearer labeling, particularly the cost data presented in logarithmic form (specify in figure legends).
Response 4.8. Figures now have all the necessary labeling information, including the logarithmic presentation, which is mentioned both in the actual image and in the figure caption.
Comment 4.9. Discussion: The interpretation is rich and cites relevant literature, but the section is overly long and could be more concise.
Response 4.9. We have now removed irrelevant phrases from the discussion to make it more concise.
Comment 4.10. Discussion: It would benefit from clearer subheadings (e.g., “Sx Differences”, “MetS Prevalence”, “Policy Implications”).
Response 4.10. The subheadings have been added.
Comment 4.11. Discussion: Add a comment on how findings could influence Romanian health policy or DRG refinement.
Response 4.11. A comment on how findings could influence Romanian health policy or DRG refinement has been added.
Comment 4.12. Limitations: Well acknowledged but should better distinguish between design-based limitations (e.g., retrospective design) and data-based limitations (e.g., ICD coding bias).
Response 4.12. This distinction has now been clearly stated in the limitations section.
